# Effect of Aging of Orthodontic Aligners in Different Storage Media on Force and Torque Generation: An In Vitro Study

Tarek M. Elshazly [1,*,†] , Diva Nang [1,†], Bijan Golkhani [1], Hanaa Elattar [2,3] and Christoph Bourauel [1]

1   Oral Technology, Dental School, University Hospital Bonn, 53111 Bonn, Germany
2   Orthodontic Department, Faculty of Dentistry, Umm Al-Qura University, Makkah 24382, Saudi Arabia
3   Orthodontic Department, Faculty of Dentistry, Suez Canal University, Ismailia 41522, Egypt
*   Correspondence: tarek.m.elshazly@gmail.com; Tel.: +49-176-344-52457
†   These authors contributed equally to this work.

**Highlights:**

- Aging effect of saliva on mechanical properties of aligners is insignificant.
- In artificial aging, artificial saliva or deionized water have similar effect.
- The experimental studies of aligners have limitations.
- Force generation by aligner is direction dependent.

**Abstract:** The aim of this study is to study the effect of aging in different media (deionized water and artificial saliva) on the force/torque generation by thermoplastic orthodontic aligners. Ten thermoformed aligners, made of Essix ACE® thermoplastic sheets, were aged in deionized water and in artificial saliva over two weeks at 37 °C, five in each medium. The force/torque generated on upper second premolar (Tooth 25) of a resin model was measured at day 0 (before aging), 2, 4, 6, 10, and 14, using a biomechanical test set-up. The results showed that aging of aligners by storage in aging agent has no significant impact on their force/torque decay. No significant differences were also found in force/torque between the aligners stored in deionized water nor artificial saliva. The vertical extrusion-intrusion forces were measured in the range of 1.4 to 4.6 N, the horizontal oro-vestibular forces were 1.3 to 2.5 N, while the torques on mesio-distal rotation were 5.4 to 41.7 Nmm. It could be concluded that the influence of saliva only on the mechanical properties can be classified as insignificant, and no significant difference between artificial aging in deionized water or artificial saliva was observed.

**Keywords:** biomechanics; orthodontic force; torque; tooth movement; removable thermoplastic appliance; aging; aligners

## 1. Introduction

In the last two decades, orthodontic aligners have become very popular among patients due to their transparency, making them a more cosmetically appealing option for adults and older teens who are self-conscious about wearing traditional metal braces. Furthermore, they can be removed for eating, brushing, and flossing, making it easier to maintain good oral hygiene. Moreover, they do not have metal brackets or wires that can cause discomfort or irritation in the mouth, making them comfortable and are preferred to fixed orthodontic braces [1–3]. Initially, the fabrication of aligners was done manually, which took a lot of effort and time. Thanks to CAD/CAM and intraoral scanner technology, the workflow has become much easier and faster [4]. The mechanical performance of orthodontic aligners is still vague and not clearly understandable [5–7]. It is assumed that the returning forces of the aligners on the target tooth are generated by two mechanisms: initially by the local elastic deformation of the aligner body in the contact area with the misaligned target tooth and then comes the elastic deformation of the entire aligner, since the entire aligner

body is lifted in the area of the misaligned target tooth while it is fixed to the other teeth by friction [8,9]. Elshazly at al. [6], based on their finite element study, postulated that the deformation of the aligner is a combination of stretching (direct proportional with deflection) and bending (direct proportional to the third power of the thickness).

Orthodontic treatment can be successful with different types of materials. However, the choice of material has a strong effect on the performance of orthodontic aligners and it depends on the specific needs and preferences of the patient and the orthodontist's recommendations [5,6]. Ideally, the orthodontic material should deliver a continuous light force throughout the treatment period [10,11]. In addition, the material should return to its original shape after removal from the oral cavity. For this purpose, the material should have sufficient stiffness to apply the required light force and a sufficiently high elastic limit to avoid permanent deformation [12]. However, the commonly used aligner sheets are made of viscoelastic thermoplastic polymers [4,7,13]. Viscoelastic materials exhibit both viscous (fluid-like) and elastic (solid-like) behavior when they are deformed. They show phenomena of creep and stress relaxation, which means that their mechanical behavior differs significantly over time under different loading conditions [12,14].

The aligners are supplied to patients in the form of a set of plastic splints. Patients are instructed to wear each aligner for at least 22 h per day for about two weeks and then replace it with its successor [15]. This means that the aligners should only be removed when eating or brushing teeth [4–7]. During function in mouth, the aligners are subjected to thermal and mechanical alternating loads. In some experimental studies [16–20] it was determined that both mechanical pre-loading and thermocycling significantly affect the mechanical properties of aligner materials, resulting in an increase in Vickers hardness [16–18], a decrease in the modulus of elasticity, as well as a reduction in strength [19,20]. Other studies [14,15,21] reported that aligner materials show a significant force-drop at the first few hours of use. Drake et al. [22], in an in-vivo study, reported that the tooth movement by aligners occurs mainly within the first week of use, thus the intraoral aging of the aligner materials has an insignificant effect on the magnitude of tooth movement.

Therefore, clinical aspects should be considered when investigating the material behavior of orthodontic aligners, and either an in vivo study should be conducted or the clinical situation should be mimicked in vitro [23]. In the literature, different media have been experimentally used to study aging of aligner materials. To our knowledge, no study has investigated whether the type of used aging medium can affect the mechanical behavior of the material. Therefore, the aim of the present experiment was to investigate the influence of different aging media (deionized water and artificial saliva) on the force/torque generation by thermoformed orthodontic aligners made of thermoplastic sheets.

## 2. Materials and Methods

Ten thermoplastic sheets of Essix ACE® (Dentsply Sirona, Bensheim, Germany) were utilized. The aligners were fabricated by thermoforming process (according to the manufacturer's instructions), using a standard thermoforming device (Biostar®; Scheu-Dental, Iserlohn, Germany), over an aligned maxillary arch model made of a fast-cold-curing resin (Technovit 4004; Kulzer, Hanau, Germany). The Biostar is a device that uses infrared heating to shape materials under controlled temperature and pressure. It can be programmed with specific parameters for heating and cooling, and is used to create aligners for teeth. The process starts by placing a 3D printed model on the platform, and a sheet of material in a clamping frame. The device is then activated, and the heating unit is positioned above the material, activating the heating process. Once the programmed heating time has elapsed, the heating unit is turned off, the chamber is positioned over the model, and air pressure is applied. After the programmed cooling time, the chamber is unlocked and the formed aligner is removed for cutting and trimming. The thermoformed aligners were then trimmed in a scalloped form following the gingival margin (Figure 1). As all aligners were made identically on the same model and by a solo well-trained technician, the effect of the aligner geometry on the measurements was negligible.

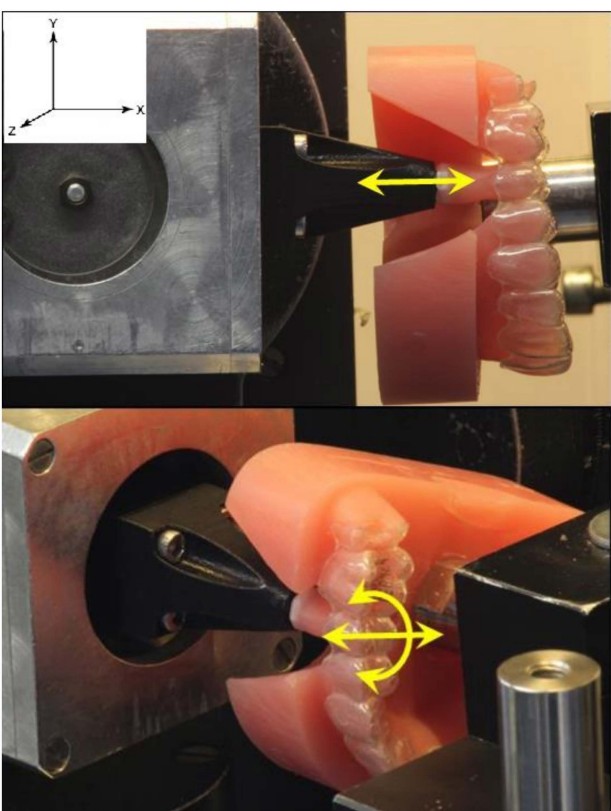

**Figure 1.** Resin model with aligner fitted in the orthodontic measurement and simulation system (OMSS): The tooth to be moved (Tooth 25) was adjusted in the aligned position in the aligner. Simulation of three tooth movements: intrusion/extrusion by 0.2 mm (X axes) (upper), oro-vestibular translation by 0.2 mm (Z axes), and 2° rotation around the tooth axis in disto-mesial direction (around X axes) (lower).

To study the effect of storage medium at mouth temperature (37 °C), five aligners were stored in a container with deionized water (Ampuwa; Fresenius Kabi, Bad Homburg, Germany) and another five in a container with modified artificial saliva. The composition of the artificial saliva was prepared according to Fusayama by Geis–Gerstorfer and Weber [24] (Table 1).

**Table 1.** Composition of artificial saliva according to Geis–Gerstorfer and Weber [24].

| Component | Molecular Formula | Concentration (mg/L) |
|---|---|---|
| Sodium Chloride | NaCl | 400 |
| Potassium Chloride | KCl | 400 |
| Calcium Chloride Dihydrate | $CaCl_2 \cdot 2H_2O$ | 795 |
| Sodium Hydrogen Phosphate 1-Hydrate | $NaH_2PO_4 \cdot H_2O$ | 690 |
| Potassium Rhodanide | KSCN | 300 |
| Sodium Sulfide | $Na_2S \cdot 9H_2O$ | 1.67 [normal with $5H_2O$] |
| Urea | $CH_4N_2O$ | 1000 |

Force/torque measurements were carried out using the Orthodontic Measurement and Simulation System (OMSS) (Figure 1) [7,25–27]. The OMSS is a device used to measure biomechanical characteristics, developed by the Oral Technology team at the University Hospital Bonn in Germany [25,26]. It is a software-connected mechanical system that is used to perform virtual treatment planning and simulation for orthodontic movements. It consists of two sensors that can measure forces and moments in all three planes of movement, mounted on motorized stages with full three-dimensional mobility. The device

is housed in a temperature-controlled chamber and is controlled by a computer program that can perform various types of measurements, including absolute measurements and simulations of orthodontic tooth movement. The OMSS can be used to analyze orthodontic problems statically and dynamically at the level of a two-tooth model. Overall, OMSS is a powerful tool for orthodontists and can improve the accuracy and efficiency of orthodontic treatment.

Measurements were conducted before starting storage (Day 0) as a control sample, and on the 2nd, 4th, 6th, 10th, and 14th day (±2 h) after starting storage. The OMSS is able to simulate three-dimensional tooth movements and measure force/deflection or torque/rotation ratios. For OMSS measurement, an additional resin replica model was created, in which the upper left second premolar (Tooth 25) was separated and its neighboring teeth were slightly ground proximally to ensure that the tooth was inserted without resistance. Tooth 25 was then connected to one of the two force/torque sensor of the OMSS, while the resin model was fixed to the second sensor (Figure 1). The occlusal plane of the resin model was adjusted parallel to the sensor Y- and Z-axes. The tooth was adjusted in the dental arch to be in its aligned neutral position, to ensure that no forces and/or torques were delivered by the aligner to the tooth in the starting position.

The tooth was moved by ±0.2 mm in 0.01 mm increments in the X (intrusion/extrusion) and the Z (oro-vestibular translation) directions, and by ±2° rotated in disto-mesial direction in 0.1° increments around the X axis (rotation around the tooth axis), where positive (+) represents extrusion, vestibular, mesial directions (Figure 1). After each measurement, the adjustment of the measurement setup was checked and corrected if necessary.

For each measurement, the aligners were stored in the container and only taken out shortly before assembly in the OMSS, and then were returned to this container immediately after the measurements were completed. The measurements of five aligners could last around one hour. For each measurement, torque/rotation and force/translation curves were recorded and processed for subsequent data analysis.

*Statistical Analysis*

The sample size calculation was made based on a study by Hahn et al. [8] and Simon et al. [27]. Based on a significance level of 0.05 and a study power of 80%, the minimum sample size was four samples. In order to achieve greater statistical power, five samples per group were used ($n = 5$).

The maximum values of force and torque were extracted for each group, and mean values and standard deviations of all values at maximum deflection were calculated. The results were presented as bar graphs to illustrate the forces and torques of the aligners in different media over aging time. The differences in the primary data were checked for significance. Since the data were normally distributed, *t*-tests were performed. A Bonferroni correction was carried out to take into account the multiple statistical significance tests with associated data. The difference was significant when the value was less than 0.05.

## 3. Results

The comparison between both storage media showed that the forces and torques transmitted by the aligners stored in distilled water and those stored in artificial saliva behaved slightly erratically over time; however, they were of a similar magnitude, and therefore no trend could be concluded nor significant differences (Figures 2–4, Table 2). The mean values of forces/torques showed no significant difference from day 0 to day 2 as well as from day 0 to day 14, except for a few groups. For the sake of clarity, only significances on days 2 and 14 are presented (Table 2). During the intrusion–extrusion movement (Tx) (Figure 2), the mean values of vertical forces generated by the aligners before storage were $-2.4 \pm 0.9$ N for intrusion and $1.5 \pm 0.5$ N for extrusion, and after storage minimum measured forces were $-2.2 \pm 0.6$ N for intrusion and $1.4 \pm 0.8$ N for extrusion. In the oro-vestibular translation (Tz) (Figure 3), the mean values of horizontal forces generated by the aligners before storage were $-1.7 \pm 0.2$ N for oral translation and

2.3 ± 0.2 N for vestibular translation, and after storage minimum measured forces were −1.3 ± 0.4 N for oral translation and 2.1 ± 0.2 N for vestibular translation. Furthermore, with the disto-mesial rotation (Rx) (Figure 4), the mean values of torque generated by the aligners before storage were −41.7 ± 11.2 Nmm for distal-rotation and 14.2 ± 4.7 Nmm for mesial-rotation, and after storage minimum measured torque were −8.1 ± 4.1 Nmm for distal-rotation and 2.0 ± 2.3 Nmm for mesial-rotation.

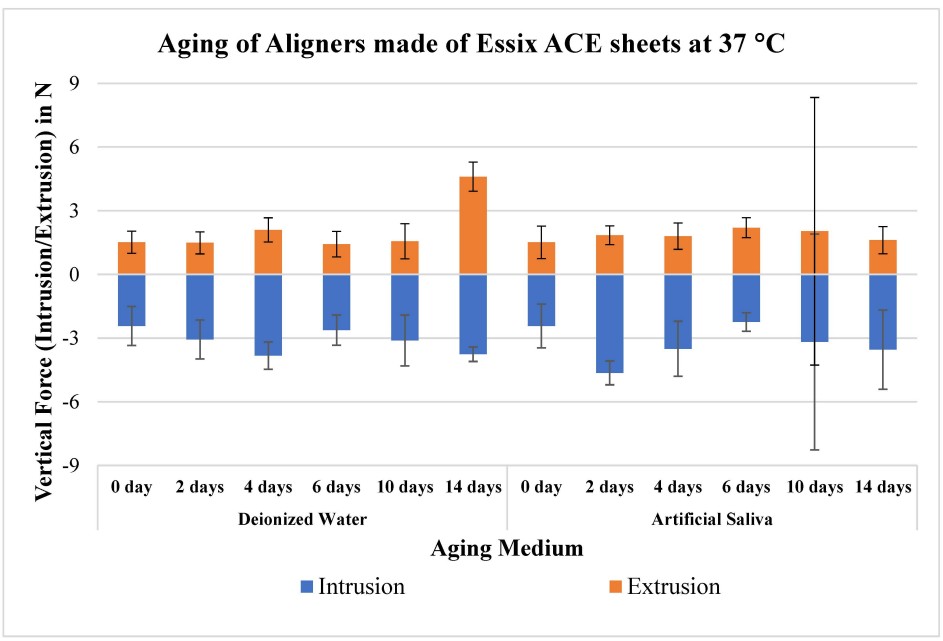

**Figure 2.** Mean of maximum intrusive/extrusive forces on tooth 25 generated by aligners made from Essix ACE® sheets after aging in deionized water and in artificial saliva at 37 °C.

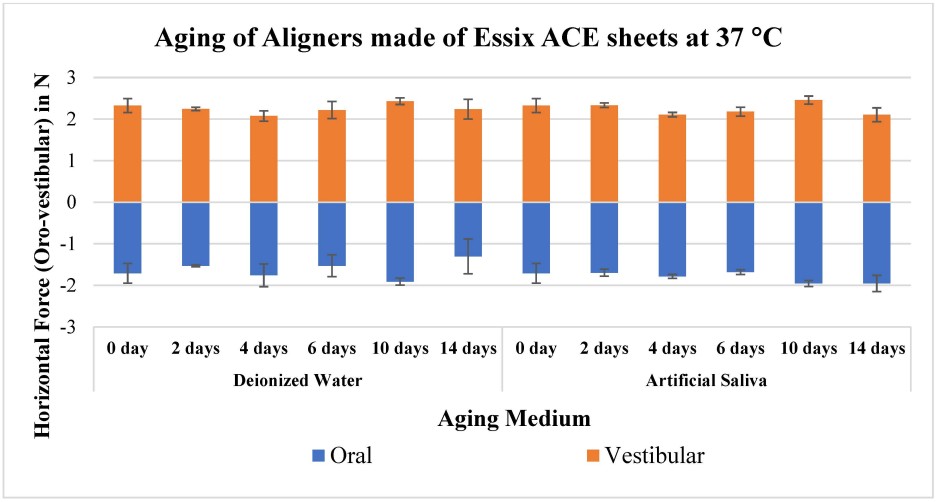

**Figure 3.** Mean of maximum oro-vestibular forces on tooth 25 generated by aligners made from Essix ACE® sheets after aging in deionized water and in artificial saliva at 37 °C.

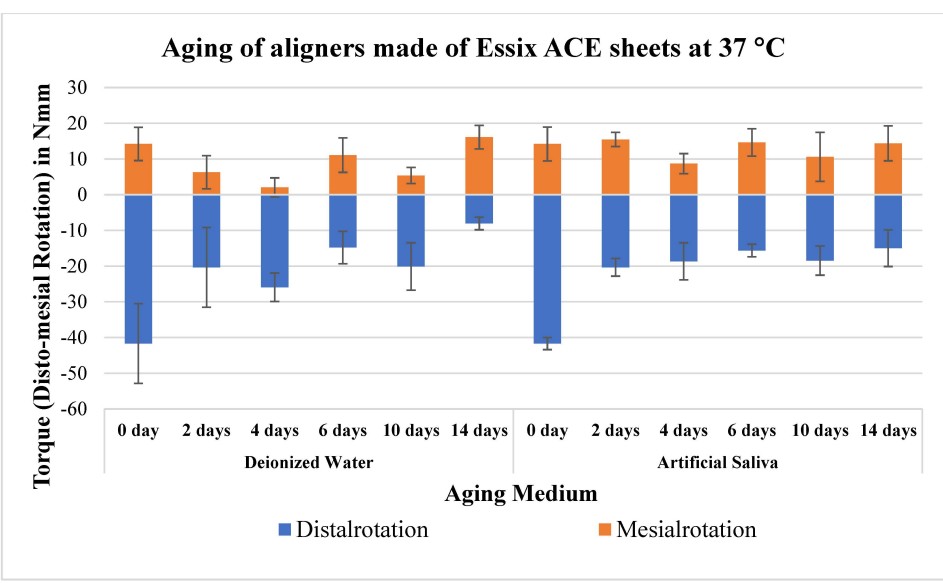

**Figure 4.** Mean of maximum torque upon rotation (disto-mesial rotation) around tooth 25 axes generated by aligners made from Essix ACE® sheets after aging in deionized water and in artificial saliva at 37 °C.

**Table 2.** Mean of maximum forces and torques on tooth 25 by aligners made from Essix ACE® sheets before and after aging (2 and 14 days) in deionized water compared to storage in artificial saliva.

| | Intrusion Force (N) | | *p*-Value | Extrusion Force (N) | | *p*-Value |
|---|---|---|---|---|---|---|
| **Aging** | **Deionized Water** | **Artificial Saliva** | | **Deionized Water** | **Artificial Saliva** | |
| **0 day** | $-2.4 \pm 0.9$ [A] | $-2.4 \pm 0.9$ [B] | - | $1.5 \pm 0.5$ [A] | $1.5 \pm 0.5$ [A] | - |
| **2 days** | $-3.1 \pm 0.6$ [Ab] | $-4.6 \pm 0.7$ [Aa] | * $p < 0.05$ | $1.5 \pm 0.6$ [Aa] | $1.9 \pm 0.6$ [Aa] | 0.121 |
| **14 days** | $-3.8 \pm 5.1$ [Aa] | $-3.6 \pm 1.9$ [ABa] | 0.899 | $4.6 \pm 6.3$ [Aa] | $1.6 \pm 0.6$ [Aa] | 0.322 |
| ***p*-value** | 0.899 | * $p < 0.05$ | | 0.899 | 0.899 | |
| | **Oral Force (N)** | | ***p*-Value** | **Vestibular Force (N)** | | ***p*-Value** |
| **Aging** | **Deionized Water** | **Artificial Saliva** | | **Deionized Water** | **Artificial Saliva** | |
| **0 day** | $-1.7 \pm 0.2$ [A] | $-1.7 \pm 0.2$ [A] | - | $2.3 \pm 0.2$ [A] | $2.3 \pm 0.2$ [A] | - |
| **2 days** | $-1.5 \pm 0.0$ [Ab] | $-1.7 \pm 0.1$ [Aa] | * $p < 0.05$ | $2.2 \pm 0.0$ [Ab] | $2.3 \pm 0.1$ [ABa] | * $p < 0.05$ |
| **14 days** | $-1.3 \pm 0.4$ [Ab] | $-2.0 \pm 0.2$ [Aa] | * $p < 0.05$ | $2.2 \pm 0.2$ [Aa] | $2.1 \pm 0.2$ [Ba] | 0.336 |
| ***p*-value** | 0.899 | 0.899 | | 0.899 | * $p < 0.05$ | |
| | **Distalrotation Torque (Nmm)** | | ***p*-Value** | **Mesialrotation Torque (Nmm)** | | ***p*-Value** |
| **Aging** | **Deionized Water** | **Artificial Saliva** | | **Deionized Water** | **Artificial Saliva** | |
| **0 day** | $-41.7 \pm 11.2$ [A] | $-41.7 \pm 11.2$ [A] | - | $14.2 \pm 4.7$ [A] | $14.2 \pm 4.7$ [A] | - |
| **2 days** | $-20.4 \pm 4.0$ [Ba] | $-20.3 \pm 4.6$ [Ba] | 0.899 | $6.3 \pm 2.7$ [Bb] | $15.5 \pm 4.8$ [Aa] | * $p < 0.05$ |
| **14 days** | $-8.1 \pm 4.1$ [Bb] | $-15.0 \pm 5.1$ [Ba] | * $p < 0.05$ | $16.1 \pm 6.9$ [Aa] | $14.4 \pm 4.9$ [Aa] | 0.674 |
| ***p*-value** | * $p < 0.05$ | * $p < 0.05$ | | * $p < 0.05$ | 0.899 | |

Different uppercase and lowercase superscript letters indicate a statistically significant difference within the same vertical column and horizontal row respectively, * significant ($p < 0.05$).

## 4. Discussion

As an active orthodontic appliance, aligners should basically have the property of exerting force evenly over time [4,7,18]. However, aligners are exposed to various intraoral loads which may affect their mechanical properties and thus their effectiveness [23]. Mechanical stresses in the oral cavity include short-term insertion and removal of the aligner, as well as the long-term contact with teeth [12]. Other environmental conditions

that may affect the mechanical properties of aligners intraorally include water absorption and temperature changes [28]. Both effects can lead to a weakening of the aligner material over time, which occurs faster at 37 °C than at room temperature [14]. Furthermore, when the polymer sheets are stored in water, the water is absorbed by diffusion, allowing the water molecules to get between the polymer chains, which facilitates their movement. With increasing temperature, the mobility of the polymer molecules also increases, leading to changes in the material [20,29–31].

In the present work, aligners made of Essix ACE® sheets (copolyester of a polyethylene terephthalate (PET) with 0.75 mm thickness) were studied to investigate the influence of artificial aging in two different media on the generation of forces/torques. For that, aligners were aged over two weeks at mouth temperature (37 °C) in deionized water and in artificial saliva. The artificial saliva was prepared according to Fusayama by Geis–Gerstorfer and Weber [24], which is used primarily in corrosion test procedures for metallic dental materials [32]. However, since the oral flora contain excretory products and plaque deposits, the use of artificial saliva is only an approximation of the in vivo situation, and individual factors and variations in saliva composition are not taken into account [32,33]. The current results suggest that the use of artificial saliva is not necessary in experimental studies of aging of aligners because there are no significant differences in the generation of torques/forces compared to deionized water.

To create a realistic simulation of mechanical intraoral conditions, the Orthodontic Measurement and Simulation System (OMSS) was used, which is equipped with a temperature-controlled chamber, whose temperature was set at 37 °C. With the help of its force/torque sensors and in conjunction with the three-dimensional variable positioning tables and the corresponding software, it allows the dynamic acquisition of the orthodontic tooth movement [25,26]. However, the rigid connection and the absence of the periodontal ligament (PDL) are a limitation of this technique.

The results of the current study showed that the vertical intrusion–extrusion forces were measured in the range of (1.4–4.6) N, the horizontal oro-vestibular forces were (1.3–2.5) N, while the torques on disto-mesial rotation were (5.4–41.7) Nmm. Also, the generation of forces and torques is direction-dependent, referring to the different oral and vestibular morphologies of each tooth [6].

Proffit [34] recommended forces of 0.1–0.2 N for intrusion, and 0.35–0.6 N for uncontrolled tipping and rotation. The literature does not offer any clear guidelines with regard to the torque, but one could set the values mentioned in relation to the root surface in order to get an indication of the appropriate torque for the maxillary second premolar (tooth 25), targeted tooth of the current study. Torque of 30 Nmm [35] or 20 Nmm [36] referred to a molar, which has an average root surface of about 430 mm$^2$ [37]. A root surface of 220 mm$^2$ is specified for tooth 25 [37]. Accordingly, the appropriate torque for tooth 25 is around 15.8 Nmm, which is consistent with that reported by Sander et al. [38] who measured torque during rotation of premolars and canines.

The intrusive–extrusion forces in this study averaged a maximum of 4.6 ± 0.7 N, which is up to 20 times higher than the 0.1–0.2 N recommended by Proffit. The reason why such recommended low forces are used for intrusion is that periodontal damage is most frequently detected with intrusions [34]. This may be of less relevance, since the magnitude of the force generally is postulated to play a minor role in the development of side effects, as long as the tooth deflection distance is less than or equal to the width of PDL (0.1–0.3 mm), as this would not stop the capillary blood flow [39]. The incremental deflection commonly used in aligner therapy and the displacement used in the current experimental work are both within the width of the PDL. Additionally, with aligner therapy, there are mutual forces of the tooth and aligner, which lead to a reduction in the range of movement that actually comes into play [40]. Furthermore, in vivo, the targeted tooth moves away from the starting point and thus the force initially acting on the tooth is reduced [39].

Similar to the current study, three-dimensional measurement and simulation setups have been used in various studies. Elkholy et al. investigated the forces/torques of aligners

on different teeth during different tooth movements [41–44]. They reported forces in palatal deflection of maxillary central incisor in the range of 4.5–7.2 N [41], and torques on a 10° mesial rotation at 73.6 Nmm [43], while the average torque of a mandibular canine on a 15° distal rotation was 42.5 Nmm [44]. It should be noted that the rotation in the present study was only 2°, which is one-fifth of the rotation in the study by Elkholy et al. [43]. In addition, the present aligners were artificially aged, which could also speak for the reduced force/torques in comparison. Similarly, Hahn et al. [8,9] reported horizontal forces on an upper central incisor tilted by 0.15 mm of 2.7 N for vestibular tilting and 3.1 N for oral tilting, which are also consistent with the oro-vestibular translation values of the present study, in terms of deflection (0.2 mm), aging, and material type. Likewise, Engelke et al. [45] studied the force and torque generation of aligners on a maxillary central incisor after simulated aging. Torques were measured between 4.3–20.2 Nmm, similar to the current work.

Using pressure-sensitive films, Elshazly et al. [5] reported forces of 3.8 N on a 0.2 mm bodily translation of an upper central incisor of resin model by Zendura FLX aligners trimmed in scalloped design as applied in the current study. Similarly, Barbagallo et al. [46] reported, in an in vivo study, a mean force of 5.1 N on 0.5 mm buccally tilted maxillary first premolars, exceeding Proffit's recommendations by several times. In the current study, a maximum force of $2.5 \pm 0.1$ N was measured for tooth 25 in vestibular translation, which is almost 50% of the value reported by Barbagallo. However, the maximum displacement in this work was only 0.2 mm (40% of Barbagallo), which means that the two values are in agreement.

Kwon et al. [18] carried out three-point bending tests on thermoplastic aligner sheets. For a 0.2 mm deflection distance of Essix ACE® sheets (0.76 mm thick), the generated force was 0.5 N before and after thermocycling with no significant difference, up to six times lower than those of current study results. The reason for this difference may be the shape of the specimen, since in the present experiment the sheets were pulled over an anatomical upper jaw model, while in Kwon's study they were pulled over a flat rectangular plaster model [9,44]. However, the results of Kwon et al. showed that the thermocycling of the aligners has no significant impact on their force loss, compatible with our results [18]. On the contrary, Barbagallo et al. [46] measured in an in-vivo study the forces at the beginning of the two-week treatment and at the end. They found that the average force decreased from 5.1 N initially to $-2.7$ N on the last day. It could be argued that the reason for force decay on aging is not referred to water/saliva storage (as tested in the current study) but other factors.

The limit of this in vitro work consists of the fact that the in vitro aligner is mechanically stressed only on certain days and for small fractions of time, while in clinical situation the aligners are worn for one to two weeks intraorally and undergo uninterrupted mechanical stress, thanks due to their elastic capacities they displace the tooth. Additionally, the absence of PDL in such mechanical devices is a limitation of the method. Moreover, the current study has reported on one type of aligner materials (Essix ACE®), however, the type of material has a significant effect on the force generation by the orthodontic aligner treatment [6,47]; hence, the effect of aging on different materials, including mechanical loading, should be considered and investigated in future studies.

### 5. Conclusions

- The influence of artificial saliva only (as an approximation of natural saliva) on the mechanical properties of an aligner can be classified as insignificant, but other studies on the combined effect of saliva together with the aligner deformation over the 14 days use are needed;
- No significant difference between the forces/torques of the aligners stored in distilled water and those stored in artificial saliva, which means that the effect of the type of used medium for aging of aligners in vitro is insignificant; and
- Generation of forces and torques by aligner is direction-dependent due to tooth morphology.

**Author Contributions:** Conceptualization: C.B.; Data curation and Analysis, Investigation, and Methodology: T.M.E., D.N., C.B., B.G.; Resources: C.B.; Software: Supervision, Validation and Visualization: T.M.E., C.B.; Writing—original draft: T.M.E., D.N.; Writing—review & editing: T.M.E., C.B., H.E. All authors have read and agreed to the published version of the manuscript.

**Funding:** This research received no external funding.

**Institutional Review Board Statement:** Not Applicable.

**Informed Consent Statement:** Not Applicable.

**Data Availability Statement:** On request.

**Conflicts of Interest:** The authors declare no conflict of interest.

**Compliance with Ethics Requirements:** This article does not contain any studies with human or animal subjects.

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
