# Peer review of "Effect of Aging of Orthodontic Aligners in Different Storage Media on Force and Torque Generation: An In Vitro Study"

_2673-6373, doi:10.3390/oral3010007_

Round 1

Reviewer 1 Report

Good work in vitro. Too many self-citations and dated bibliography. Prefer recent and open access papers. 1. Are few statistical data significant for justifying the results?

2. line 97-please describe what the Orthodontic Measurement and Simulation System (OMSS) is and add the supporting bibliography without self-citations. Specify the manufacturer and country.

3. Why was the premolar in position 2.5 chosen? Justify your choice
4. Why was deionized water used to validate the hypothesis of the study? What is it compared to? Wouldn't it have been better to compare a new aligner with an aligner aged in saliva that is always new and finally an aligner that has aged and worked
5. Avoid excessive self-citations and include recent work on MDPI to support the study hypothesis.

6. line 263" The comparison between the forces/torques over 14 days of the respective material showed no significant difference, meaning that the influence of saliva on the mechanical properties of an aligner in everyday clinical practice can be classified as insignificant. "
cannot be written because as already discussed above the work is in vitro and therefore not comparable to natural saliva and to use in the mouth under deformation for 14 days. Correct and clarify the appropriate conclusions

Author Response

Dear Reviewer,
We are grateful for your kind consideration and the time you dedicated to reviewing our submission. These considerations have been important for us to improve the quality of our report and are relevant for us.
We have addressed each consideration separately and submitted the revised manuscript.

Response to comments point by point:

1- Are few statistical data significant for justifying the results?

Our response:

Sample size calculations were done based on similar previous studies of force measurements of aligners by similar custom-made devices. In the previous studies, they used a sample size of 3-5 aligners per group, and here we used 5 aligners per group.

The following paragraph was added: (The sample size calculation was made based on a study of Hahn et al. [8] and Simon et al. [27]. Based on a significance level of 0.05 and a study power of 80%, the minimum sample size was four samples. In order to achieve greater statistical power, five samples per group were used.)

2- line 97-please describe what the Orthodontic Measurement and Simulation System (OMSS) is and add the supporting bibliography without self-citations. Specify the manufacturer and country.

Our response:

The following paragraph was added:

(The OMSS is a device used to measure biomechanical characteristics, developed by the Department of Oral Technology at the University Hospital Bonn in Germany [25, 26]. It consists of two sensors that can measure forces and moments in all three planes of movement, mounted on motorized stages with full three-dimensional mobility. The device is housed in a temperature-controlled chamber and is controlled by a computer program that can perform various types of measurements, including absolute measurements and simulations of orthodontic tooth movement. The OMSS can be used to analyze orthodontic problems statically and dynamically at the level of a two-tooth model.)

3. Why was the premolar in position 2.5 chosen? Justify your choice

Our response:

Actually, in our lab, we run a wide-scale project on orthodontic aligners, in which we are measuring the forces/torques generated on several teeth in several conditions by different aligner materials and designs

In the current study, we just reported on tooth 25 since to our knowledge, no study in the literature reported the effect of aging on force generation by aligners on tooth 25. Drake et al. [22] reported on tooth 11, but here we reported on tooth 25. (Introduction, lines 64-74)

4. Why was deionized water used to validate the hypothesis of the study? What is it compared to? Wouldn't it have been better to compare a new aligner with an aligner aged in saliva that is always new and finally an aligner that has aged and worked

our response: 

>> As mentioned throughout the manuscript, clinical aspects should be considered when investigating the material behavior of orthodontic aligners, and either an in vivo study should be conducted or the clinical situation should be mimicked in vitro.

In the literature, several studies reported on the aging of aligners either by using deionized water or artificial saliva, here we tried to report if there is a significant difference when using any of the aging agents or not! especially since artificial saliva is not always available or easy to be prepared.

>> The aging on day 0 is considered a control (new aligner without aging)

>> In our coming study we are investigating the combined effect of mechanical deformation together with aging, here we wanted only to compare the effect of using both aging agents. Our initial results showed the significant effect of mechanical deformation over the 14 days on the forces/torque generation.

5. Avoid excessive self-citations and include recent work on MDPI to support the study hypothesis.

Our response:

We thought that the review process is blind

The self-citation is only used because as mentioned we run a wide-scale project in aligners and it doesn't affect the quality of the results and we can't be considered as a bias. Our other studies are about using smart polymers in aligners, finite element simulations of aligners, polymer aging, and force generation by aligners using our custom-made device (OMSS) which is well-known in the literature.

6. line 263" The comparison between the forces/torques over 14 days of the respective material showed no significant difference, meaning that the influence of saliva on the mechanical properties of an aligner in everyday clinical practice can be classified as insignificant. " cannot be written because as already discussed above the work is in vitro and therefore not comparable to natural saliva and to use in the mouth under deformation for 14 days. Correct and clarify the appropriate conclusions

Our response:

Thanks for the correction

The paragraph was corrected as follows:

(The influence of aging in artificial saliva only (as an approximation of natural saliva) on the mechanical properties of an aligner can be classified as insignificant, but other studies on the combined effect of saliva together with the aligner deformation over the 14 days of aligner use are needed)

Reviewer 2 Report

  I would like to congratulate the authors on their choice of theme. I also believe that the role of the reviewer is to ensure the scientific quality of the work, as well as help authors, namely to improve it as much as possible. And my contributions are in this direction.

In this sense, I have a few comments: 

The authors need  to present how they calculate the sample size for this research. Figure 4 should be presented differently. Namely, at the same time, samples from artificial saliva and deionized water should be next to each other for both mesial rotation and distal rotation, in order for the reader to compare them more easily.

Author Response

Dear Reviewer,
We are grateful for your kind comments and the time you dedicated to reviewing our submission.
We have calculated the sample size and submitted the revised manuscript after consideration of other reviewers' comments as well.

Added paragraph:

(The sample size calculation was made based on a study by Hahn et al. [8] and Simon et al. [27]. Based on a significance level of 0.05 and a study power of 80%, the minimum sample size was four samples. In order to achieve greater statistical power, five samples per group were used.)

Round 2

Reviewer 1 Report

The limit of this in vitro work consists in the fact that the in vitro aligner is mechanically stressed only on certain days and for small fractions of time, while in reality the aligners worn for 7-14 days undergo uninterrupted mechanical stress by virtue of which, thanks due to their elastic capacities they displace the tooth. This limitation should be thoroughly clarified in the presentation of the abstract and in the discussion. Please reduce self-citations and include recent and open access bibliography. The peer review process is blind on your part but the reviewers read the authors name. Good work

Author Response

Thanks a lot again for your thorough review and for your time

The comment was added to the end of the discussion as a limitation of the study